# Biological Properties and Prospects for the Application of Eugenol—A Review

**DOI:** 10.3390/ijms22073671

**Published:** 2021-04-01

**Authors:** Magdalena Ulanowska, Beata Olas

**Affiliations:** Department of General Biochemistry, University of Lodz, Pomorska 141/3, 90-236 Lodz, Poland; magdalena.ulanowska@edu.uni.lodz.pl

**Keywords:** eugenol, clove oil, antibacterial, antifungal, antioxidant

## Abstract

Eugenol is a phenolic aromatic compound obtained mainly from clove oil. Due to its known antibacterial, antiviral, antifungal, anticancer, anti-inflammatory and antioxidant properties, it has long been used in various areas, such as cosmetology, medicine, and pharmacology. However, high concentrations can be toxic. A dose of 2.5 mg/kg body weight is regarded as safe. This paper reviews the current state of knowledge regarding the activities and application of eugenol and its derivatives and recent research of these compounds. This review is based on information concerning eugenol characteristics and recent research from articles in PubMed. Eugenol remains of great interest to researchers, since its multidirectional action allows it to be a potential component of drugs and other products with therapeutic potential against a range of diseases.

## 1. Introduction

Eugenol is a well-known and well-studied compound. It was first isolated in 1929 as a volatile compound from *Eugenia caryophyllata*, and commercial production began in the USA in 1940 [1,2].

Due to its numerous properties, eugenol has found a wide range of uses in many areas of life. In addition, because of the growing interest in unconventional, traditional medicines that contain natural ingredients, it remains an important object of scientific research as a potential component of various medicinal products, including those intended for treating human lung cancer [3,4]. Isoeugenol derivatives have become a popular subject of research due to their fungicidal and insecticidal properties, because they exhibit greater antimicrobial activity than eugenol [2]. This study reviews the current state of knowledge regarding the multidirectional action and application of eugenol and its derivatives and recent research into these compounds.

## 2. Characteristics of Eugenol

Eugenol (C_10_H_12_O_2_; phenylpropanoid, Figure 1) is an aromatic compound belonging to the group of phenols. It is commonly obtained from the natural essential oils of plants from the *Lamiaceae*, *Lauraceae*, *Myrtaceae* and *Myristicaceae* families, and is the most important component of clove oil (*Syzygium aromaticum*). Although it is known to occur in various concentrations depending on the species (Table 1), the richest source is *S. aromaticum*, where it constitutes between 9381.7 mg and 14,650 mg per 100 g of fresh plant material and is primarily responsible for its characteristic aroma [2,5,6,7,8].

Eugenol is a clear to pale yellow liquid with an oily consistency and a spicy aroma. It is sparingly soluble in water and well soluble in organic solvents. Eugenol can be produced synthetically in two ways, one of which involves the allylation of guaiacol with allyl chloride. The biotechnological method is based on the biotransformation of a wide range of microorganisms, such as *Corynebacterium* spp., *Streptomyces* spp., and *Escherichia coli* [2,6,7,8].

Eugenol has low chemical stability and is sensitive to oxidation and various chemical interactions. When orally administered, it is rapidly absorbed by various organs and metabolized in the liver. Therefore, encapsulation of eugenol seems to be the best solution to prevent early absorption, improve its water solubility, and, thus, increase its activity (e.g., it has been reported that the amount of eugenol delivered increases at least sixfold in infected cells when delivered as solid lipid nanoparticles [2]). The eugenol inclusion complexes may have enhanced thermal stability and, as a result, provide slow release of eugenol. These may be microemulsions containing eugenol prepared by simply dissolving the eugenol (0.75–1.5% *w*/*w*) essential oil in micelles of surfactants [2,9].

Eugenol has demonstrated various antioxidant, analgesic, antimutagenic, anti-platelet, antiallergic, anti-swelling, and anti-inflammatory properties. It has also displayed antimicrobial effects against many human pathogens, including a wide group of Gram-positive and Gram-negative bacteria and fungi and a number of parasites, including *Giardia lamblia, Fasciola gigantica*, and *Haemonchus contortus*. Furthermore, eugenol can protect against carbon tetrachloride (CCl_4_)-induced hepatotoxicity [2,4,5,7,11].

Despite many important properties, eugenol can cause irritation and allergy. There have been reports of allergic contact dermatitis, presenting as hand eczema in dentistry workers like dentists or dental assistants (due to the use of eugenol in dentistry), as well as allergic contact gingivitis or burning mouth syndrome [12].

## 3. Antioxidant Activity

Eugenol is a popular antioxidant and monoamine oxidase (MAO) inhibitor, and it is also known to exhibit neuroprotective properties [13]. Eugenol is known to scavenge free radicals, inhibit the generation of reactive oxygen species, prevent the production of reactive forms of nitrogen, increase cyto-antioxidant potential, and protect the function of microbial DNA and proteins. Eugenol can also help repair oxidative damage, eliminate damaged molecules, and prevent mutations that could develop into cancer [1,7,14]. The antioxidative potential of eugenol has been attributed to its structure, which allows it to fix phenoxy radicals by receiving donated hydrogen atoms [15].

Both clove oil and eugenol demonstrate strong antioxidant potential. Pérez-Rosés et al. [16] found both to have a strong DPPH (2,2-diphenyl-1-picrylhydrazyl) radical scavenging effect (half maximal inhibitory concentration (IC_50_) = 13.2 μg/mL for clove oil; 11.7 μg/mL for eugenol) and to inhibit reactive oxygen species (ROS) production in human neutrophils stimulated by phorbol 12-myristate 13-acetate (IC_50_ = 7.5 μg/mL for clove oil; 1.6 μg/mL for eugenol) or H_2_O_2_ (IC_50_ = 22.6 μg/mL for clove oil; 27.1 μg/mL for eugenol). They also inhibit the production of nitric oxide (IC_50_ = 39.8 μg/mL for clove oil; 19.0 μg/mL for eugenol) and demonstrate high myeloperoxidase (MPO) inhibition in human leukocytes (IC_50_ = 16.3 μg/mL for clove oil; 19.2 μg/mL for eugenol).

While eugenol is known to have antioxidant and anti-inflammatory properties at low doses, a pro-oxidative effect can occur at higher concentrations, resulting in the formation of free radicals. In addition, many studies have shown that the administration of high concentrations of clove oil can increase the number of DNA breaks in normal human fibroblast cells [1,8,10,17].

## 4. Antimicrobial Activity of Eugenol

Eugenol has demonstrated antibacterial properties against many species, such as *Staphylococcus aureus*, *Pseudomonas aeruginosa*, and *Escherichia coli*, and this potential has been attributed to the free OH group in its structure. Against Gram-negative bacteria, eugenol is believed to act by damaging the cytoplasmic membrane; being a hydrophobic molecule, it can easily penetrate the lipopolysaccharide cell membrane and enter the cytoplasm. Once present in the cell, it can cause alterations to the cell structure, resulting in the leakage of intracellular components.

In *Enterobacter aerogenes*, it has been proposed that the hydroxyl group on eugenol inhibits the action of protease, histidine carboxylase, and amylase by binding to them ([2,18]). Similarly, eugenol has been found to potentially inhibit the activity of membrane-bound ATPase in *Escherichia coli* and *Listeria monocytogenes* [19]. It has also demonstrated synergistic effects with conventional antimicrobials [9]. It is further believed that eugenol is capable of producing intracellular reactive oxygen species (ROS), which can cause cell death by inhibiting cell growth, disrupting the cell membrane, and damaging DNA [20].

Eugenol was found to inhibit the growth of human isolates of *Streptococcus agalactiae* (planktonic GBS—group B streptococci strain), including those resistant to erythromycin and clindamycin [21]. After a five-hour incubation with 0.125% to 0.5% eugenol, the GBS planktonic cells demonstrated leakage of proteins and lipids from the cytoplasm and disruption of the cell membrane.

Da Silva et al. [11] also report that eugenol derivatives demonstrated a higher antimicrobial potential than eugenol: the minimum inhibitory concentration (MIC) against bacteria was found to be 500 µg/mL for the derivatives and 1000 µg/mL for eugenol. The derivatives were formed by the esterification of the hydroxyl group (-OH) with various carboxylic acid derivatives or by the addition of functional groups to the double bond of the allyl group.

In addition to its significant antibacterial properties, eugenol has also demonstrated antiviral activity; it acts synergistically with acyclovir in inhibiting the herpes virus in vitro and against HSV-1 and HSV-2 (*Herpes simplex virus* 1/2) by preventing viral replication and limiting viral infection. Eugenol has also been found to possess antifungal activity against a range of fungal strains in vitro, including *Candida albicans*, *Aspergillus niger*, *Penicillum glabrum*, *Penicillum italicum*, *Fusaria oxysporum*, *Saccharomyces cerevisiae*, *Trichophyton mentagrophytes*, *Lenzites betulina*, *Laetiporus sulphurous*, and *Trichophyton rubrum*. In the case of fungi, it is thought that eugenol disturbs cell membrane function, inhibits virulence factors, and prevents fungal biofilm formation [2,4,9]. The effect of eugenol against bacteria and fungi is shown in Figure 2.

Sharifzadeh and Shokri [22] investigated the antifungal potential of eugenol using the broth microdilution test and the likely synergistic effect of eugenol with voriconazole in vitro against *Candida* strains isolated from mares’ reproductive tract using the checkerboard microdilution method. Eugenol MIC values for eugenol were 400–800 µg/mL for *Candida tropicalis* and 200–400 µg/mL for *Candida krusei*. Synergistic effects of eugenol and voriconazole were observed for *Candida tropicalis* (83.3%) and *Candida krusei* (77.7%), and no antagonistic activity occurred. Consequently, eugenol is a potential antifungal agent designed to fight the genital *Candida* yeast. Moreover, the combination therapy of eugenol and voriconazole may prove effective in antimicrobial resistance in mares with genital candidiasis.

The study on eugenol showed that this compound has synergistic activity with various antibiotics, such as vancomycin, penicillin, ampicillin, and erythromycin, and the combination of these compounds allowed a reduction in MIC values of 5–1000 times compared to the MIC values of individual compounds used alone. Moreover, eugenol has been shown to potentiate the action of lysozyme and SDS, which are used to damage bacterial cell membranes. Mass administration of antibiotics is a serious global problem, as it promotes the spread of antibiotic-resistant strains of pathogens, so limiting their use to the natural compound as eugenol seems to be a potential solution [23].

Eugenol, a component of essential oil, in addition to other compounds, such as trans-cinnamaldehyde, citronellol, and terpineol, has been tested for its effectiveness in eliminating bacterial biofilm [24]. Bacterial biofilm is a three-dimensional macrocolony of bacteria isolated by an extracellular matrix, produced by these microorganisms [25]. Bacterial biofilms are a significant problem. For example, in the food industry, they form on the surface of food products, contaminating the food and can causing disease development. Therefore, effective measures are being sought to solve this issue [24].

Olszewska et al. [24] examined the above-mentioned compounds, which are components of essential oils, in regard to their ability to inhibit the growth of the *Escherichia coli* biofilm. The study was performed on the basis of the platelet count, resazurin test, and Syto^®^ 9/PI (-/propidium iodide) staining in combination with flow cytometry (FCM) and confocal laser scanning microscopy (CLSM). Antibiofilm tests–resazurin assay and plate counts showed that eugenol at a concentration of 3 mM caused a significant decrease in the metabolic activity of bacterial cells, which are part of the biofilm (49%), and in culturability (84%), but compared to other compounds tested, eugenol showed the lowest ability to damage the microbial cell membrane.

The biofilm formed by bacteria on medical implants and biomaterials is also a common source of bacterial infection. Studies have found that a hydrophilic copolymer system based on eugenol effectively inhibited the growth of such bacteria. The general mechanism of action of eugenol on bacterial biofilm includes inhibition of biofilm formation and reduced viability of biofilm-forming cells. Other effects included dispersion of cells in the biofilm matrix, inactivation of biofilm bacterial cells, and inhibition of biofilm-associated gene expression (for example, the pgaA gene) [26]. Eugenol may also inhibit the production of bacterial virulence factors, such as violacein, elastase, and pyocyanin, and prevent biofilm formation. It also appears effective against multi-resistant strains, such as *Salmonella enteritidis* [9].

Qian et al. [26] also showed antimicrobial activity against carbapenem-resistant *Klebsiella pneumoniae* (CRKP). *Klebsiella pneumoniae* is a very dangerous pathogen that poses a risk to humans and animals due to its resistance to antimicrobial agents and antibiotics. This bacterium exhibits many virulence factors, e.g., the ability to create biofilms or the presence of capsular polysaccharide and outer membrane proteins. Carbapenem-resistant CRKP strains are a particularly significant threat, so it is so important to find an effective agent to combat or weaken this pathogen. Eugenol seems to be a potential compound showing antibacterial activity against this microorganism, as it has demonstrated multidirectional activity. Minimal inhibitory concentration of eugenol was determined by the agar dilution method, and the MIC against the four tested CRKP isolates was 0.2 mg/mL for eugenol. Importantly, a certain relationship was noticed—with the increase in eugenol concentration, the degree of cell damage and the number of damaged cells increased. The antimicrobial mechanism of eugenol was damage to the cell membrane: disruption of the cell membrane and swelling of the cells, cell membrane hyperpolarization and enhanced membrane permeability, and, finally, leakage of intracellular components of CRKP cells.

## 5. Anticancer Activity of Eugenol

Eugenol was found to induce apoptosis in human promyelocytic leukemia cells (HL-60) through a mechanism dependent on ROS and mitochondria, suggesting that it could have apoptosis-inducing chemotactic properties [27].

Evidence suggests that eugenol can affect cancer cells as an antioxidant, preventing mutation, and as a pro-oxidant, influencing signal pathways and killing cancer cells. The molecular mechanism is believed to include various stages: inhibiting NF-κB activation, downregulating prostaglandin synthesis, reducing cyclooxygenase-2 activity, inducing cell cycle arrest in the S phase, and causing apoptotic cell death by lowering inflammatory cytokine levels [3,17]. Fangjun and Zahijia [3] suggest that eugenol may have chemotherapeutic properties against human lung cancer. An in vitro study conducted on human embryonic lung fibroblast MRC-5 and lung cancer adenocarcinoma cells A549 found that even a low dose of eugenol interfered with the migration and invasion of carcinogenic cells, inhibited lung cancer cell viability, and prevented metastasis by blocking the PI3K/Akt pathway (an intracellular signaling pathway involved in cell cycle regulation) and inhibiting MMP (matrix metalloproteinase) activity. The compound was also found to demonstrate cytotoxic effects against normal cells and lung cancer cells at higher doses (1000 μM).

Eugenol was also found to enhance the cytotoxic and pro-apoptotic activity of cisplatin, a cytostatic drug, in both in vivo and in vitro studies on triple-negative breast tumors [28]. The addition of eugenol is believed to enhance the inhibition of breast cancer stem cells by cisplatin by inhibiting the activity of aldehyde dehydrogenases (ALDH) and ALDH-positive tumor initiating cells and enhancing NF-κB signaling pathway inhibition. These results suggest that combination therapy based on eugenol and cisplatin may be an effective therapy for triple-negative breast tumors.

Similarly, eugenol appears to increase the sensitivity of human immortal cell line from cervical cancer (HeLa cells) to cisplatin [17]. Greater inhibition was observed at all cisplatin concentrations when combined with eugenol compared to cells treated with cisplatin alone. These results again suggest that combining these drugs increases their effectiveness.

## 6. Anti-Inflammatory and Analgesic Activities of Eugenol

Eugenol is a popular painkiller and anesthetic used in dental practice. It has been found to inhibit voltage-gated sodium channels (VGSC) in the primary supply neurons of the teeth in various studies, including one based on a rat model [29,30].

Evidence suggests that eugenol has the ability to inhibit the production of superoxide anions in neutrophils by inhibiting the Raf/MEK/ERK1/2/p47-phosphorylation pathway. It is also known to be an inhibitor of pro-inflammatory mediators, including IL-1β and IL-6, tumor necrosis factor alpha (TNF-α), prostaglandin E_2_ (PGE_2_), expression of inducible oxide nitrate synthase (iNOS) and expression of cyclooxygenase-2 (COX-2), nuclear factor kappa B (NF-κB), and leukotriene C4 and 5-lipoxygenase (5-LOX) [31]. Its anti-inflammatory activity is associated with preventing neutrophil/macrophage chemotaxis and inhibiting the synthesis of inflammatory neurotransmitters, such as prostaglandins and leukotrienes; in addition, eugenol dimers have shown chemopreventive properties by inhibiting cytokine expression in macrophages (Figure 3) [4,32].

In silico bioinformatics studies by Das Chagas Pereira de Andrade and Mendes [31] found that eugenol may inhibit both COX-2 and 5-LOX. It is therefore possible that eugenol may act as an anti-inflammatory agent, thus allowing it to replace some NSAIDs (nonsteroidal anti-inflammatory drugs) in various diseases; it could also be used in the synthesis of new selective drugs to fight diseases associated with inflammatory processes, such as osteoarthritis or cancer.

## 7. Other Properties

Since ancient times, eugenol has been used for dental and oral care: it demonstrates antimicrobial activity against bacteria associated with dental caries and periodontal disease, as well as disinfectant properties, and has been found to relieve local pain, such as pulpitis and dentinal hypersensitivity, as a topical analgesic. In dentistry, it is combined with zinc oxide to form an amorphous chelate compound used to indirectly cover the pulp, dress endodontic treatment, and temporarily fill cavities. In liquid form, it is also used in special pastes such as mummification pastes (e.g., Caryosan and Endomethazone) to fill root canals. Furthermore, eugenol is sometimes rubbed on the gums to numb them before dentures are inserted [1,5,14,32].

Since eugenol is considered a generally safe compound at low concentrations and has multidirectional action, it is commonly used in pharmaceuticals, food, cosmetics, and as a local antiseptic and analgesic. It is also a common ingredient in household products, such as soaps, perfumes, skin care products, cigarettes where it is used as a flavor, and fragrance. It is also used as a preservative to protect foods from microorganisms as well as a pesticide and fumigant. The Joint Food and Agriculture Organization/WHO Expert Committee on Food Additives found that the maximum allowable daily intake of eugenol or clove oil is 2.5 mg/kg body weight for humans [1,4,8,14].

Due to its pro-health properties, eugenol is also used in the treatment of infections of the upper respiratory and gastrointestinal tracts and is used for joint pain. It is also included in various medications intended for the inflammation of the mucosa in the upper respiratory tract and for the prevention of colds. These medications are commonly administered as inhalation and aerosol therapy: for example, Amol, Aromatol, or Olbas [32].

Due to eugenol’s multidirectional activity, including antibacterial and antifungal, it has applications in agriculture and the food industry. Its beneficial effect is associated with low concentrations of effective action, which is an important advantage. Additionally, eugenol is effective against many foodborne pathogens (for example, *Salmonella typhi* and *Aspergillus ochraceus*), so its use prevents acute food poisoning. The anti-salmonella activity of eugenol includes a decrease in the permeability of the pathogen’s cell membrane, followed by ion leakage, loss of cellular content, and ultimately cell death [2,9]. Therefore, in agriculture, eugenol is used as a biocontrol agent for grains, because it has been found that eugenol may reduce contamination of organic products by *Salmonella* through inhibition of its growth in soil. Eugenol is also able to inhibit the production of ochratoxin A by *Aspergillus ochraceus.* Its antifungal properties are harnessed to protect fruits, such as strawberries, apples, and peaches, and their juices against the harmful effects of microorganisms (Table 2). One of the most common foodborne pathogens is *Staphylococcus aureus*. Studies suggest that the effects of eugenol on *S. aureus* are based on its ability to inhibit the production of staphylococcal enterotoxin A and B and toxic shock syndrome toxin 1 as well as the expression of alpha-hemolysin [9].

## 8. Eugenol Derivatives

Due to the rapid growth of the human population that has led to an augmented need for food, it is necessary for the agricultural industry to modify methods that combat this problem. The necessity of prevention and control of plant diseases, as well as pests, is a key issue in crop protection. To date, the most commonly used strategy to combat these problems is the use of conventional pesticides, most of which are synthetic, including insecticides [33,34,35]. The intensified use of artificial pesticides leads to harmful effects on the environment, including the decrease in biodiversity and the rise in factors potentially hazardous for human health [36,37]. Thus, less harmful strategies, for instance, the use of pesticides from natural origins, must be taken into consideration.

In recent years, the use of plant essential oils and their bioactive compounds as potentially effective biopesticides gained much importance. Essential oils are now a recognized alternative to hazardous artificial pesticides in the agricultural industry [38,39,40]. These substances possess a number of advantages—efficacy, high biodegradability, low toxicity, various modes of action, and availability of source materials. However, the use of essential oils in plant protection has some limitations, mostly due to high volatility and low solubility [41].

As mentioned above, eugenol is being widely used as a biocontrol factor because of its antifungal properties. Moreover, the literature reports that eugenol is a strong insecticide, effective against a wide range of domestic arthropod pests [42,43]. Lately, it has been shown that the various structural modification of essential oils (EOs) can enhance the biocidal potential of these phytochemicals by the high increase in their activity [44,45].

In their studies, Fernandez et al. [46] investigated the efficacy of semisynthetic eugenol derivatives—such as *O*-alkylated bearing the propyl chain with hydrogen, hydroxyl, ester, chlorine, and carboxylic acid as terminals and corresponding *O*-alkylated oxiranes against the Sf9 (*Spodoptera frugiperda*) insect cell line. The insecticide activity of eugenol derivatives was also compared to a commercial synthetic insecticide, chlorpyrifos. Results showed that all eugenol derivatives that were formed by the alkylation reactions of the hydroxyl group possessed a propyl chain, with hydrogen, hydroxyl, ester, chlorine, and carboxylic acid terminals displaying higher toxicity than the pure form of eugenol. Among the oxiranes that arise from the epoxidation of alkylated derivatives, all compounds also showed higher toxic potential. The greatest efficacy was presented by two oxiranes—2-(4-(3-chloropropoxy)-3-methoxybenzyl)oxirane and ethyl 4-(2-methoxy-4-(oxiran-2-ylmethyl)phenoxy)butanoate, whose effects on the insects’ cells were nearly twofold stronger than the effect of the commercial insecticide. These two derivatives were further analyzed to investigate the effects on insects’ cells. Extended studies showed that both eugenol derivatives caused decrease in cell density compared with control cells, which corresponded with their impact on cell viability. Moreover, both derivatives exhibited changes in the chromatin structure—both chromatin condensation and nuclear fragmentation. In addition, an investigation of the impact on human cells was performed, as evaluation of potential toxic effect on humans is crucial for the development of new pesticides. Considering the usual routes of poisoning, human skin keratinocytes were chosen for the examination. Of these two derivatives, only ethyl 4-(2-methoxy-4-(oxiran-2-ylmethyl)phenoxy)butanoate showed no harmful effects on human cells while still demonstrating strong insecticidal activity, thus displaying a selective effect on insects’ cells.

Eugenol derivatives show enhanced antifungal activity compared to the pure form of this compound. In their studies, Maximino et al. [47] investigated the fungicidal effects of eugenol and its derivatives on *Fusarium solani* f. sp. *piperis*. Results showed that 2-(4-allyl-2-methoxyphenoxy)-3-chloronaphthalene-1,4-dione was the most effective eugenol derivative on the fungus viability. This high activity is probably connected to the eugenol scaffold and its structure, which contains the naphthoquinone moiety. Naphthoquinones are considered useful compounds in the development of agrochemicals due to their characteristics, such as non-toxic nature, redox properties, and antimicrobial activity [48,49,50]. Moreover, the mentioned derivative exhibited superior antifungal activity to effects performed by tebuconazole to which the investigated fungus developed resistance [51].

The antifungal activity of eugenol, its analogues, and derivatives was also confirmed in various studies. It was reported that while eugenol is almost inactive against *Saccharomyces cerevisiae*, *Candida albicans* and *Aspergillus niger*, the isoeugenol shows a moderate inhibitory effect on the same fungi. Another study showed that both eugenol and its analogues exhibit effectiveness against a variety of fungus species, for instance *Penicilium*, *Altermaria*, and *Fusarium* spp. [52]. However, studies that concentrated on other species were lacking. Olea et al. [52] performed an investigation on the effect of eugenol and its derivatives on fungus *Botrytris cinerea*, which causes gray mold disease. This species is considered a serious worldwide problem as it causes high losses in fresh fruit crops [53]. In their study, the antifungal activity was evaluated using a resistant *Botrytris cinerea* native isolate obtained from cherry fruit. The results indicated that the chemical modification of eugenol causes significant changes in the antifungal activity. In this case, modifications included changes in the existing functional groups and the addition of a nitrogen-containing group on the aromatic ring. It was established that an increase in antifungal activity and growth inhibition is connected with conjugation of the side-chain double bound with the aromatic system. Moreover, it was considered that growth inhibition caused by eugenol derivatives is probably a consequence of two different mechanisms—accumulation of compounds in the fungal membrane due to lipophilic interactions and Michael-type reactions between eugenol derivatives and components of the fungal membrane or production of reactive oxygen species (ROS) by enzymatic reduction of nitro compounds.

## 9. Conclusions

Due to its wide range of biological activities, eugenol has many applications. It is commonly found in soaps or perfumes as a fragrance; however, it is also used in medicine and pharmacology, most commonly as a local antiseptic and analgesic, and as an anti-inflammatory agent in inhalation and aerosol therapy. More importantly, it has also demonstrated therapeutic potential in drugs, including those intended to fight cancer. Eugenol also has a synergistic effect with various antibiotics, for example, vancomycin, penicillin, and erythromycin. By potentiating their action, it reduces their minimum inhibitory concentration (MIC), which has the effect of reducing antibiotic resistance among pathogens. Unfortunately, high concentrations of eugenol can be pro-oxidative and harmful, but doses below 2.5 mg/kg body weight are regarded as safe by the FAO. Moreover, eugenol may cause allergies (for instance, allergic contact dermatitis) in some cases, especially in dental workers. Eugenol derivatives are also an important group of compounds and a popular research object. Eugenol derivatives appear to be promising ingredients in pesticides, including insecticides.

## Figures and Tables

**Figure 1 ijms-22-03671-f001:**
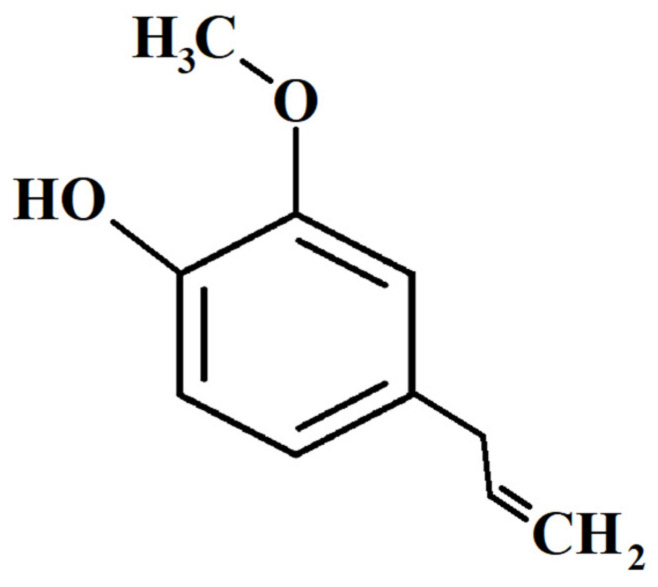
Chemical structure of eugenol.

**Figure 2 ijms-22-03671-f002:**
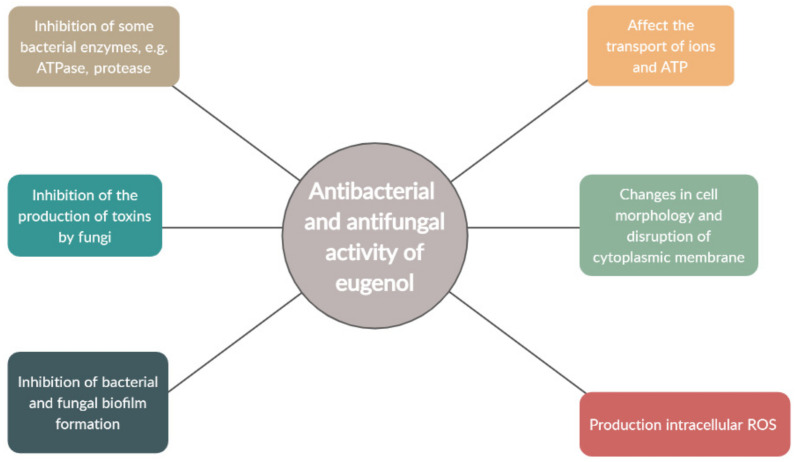
Antibacterial and antifungal activity of eugenol (own elaboration based on [2,9,20]).

**Figure 3 ijms-22-03671-f003:**
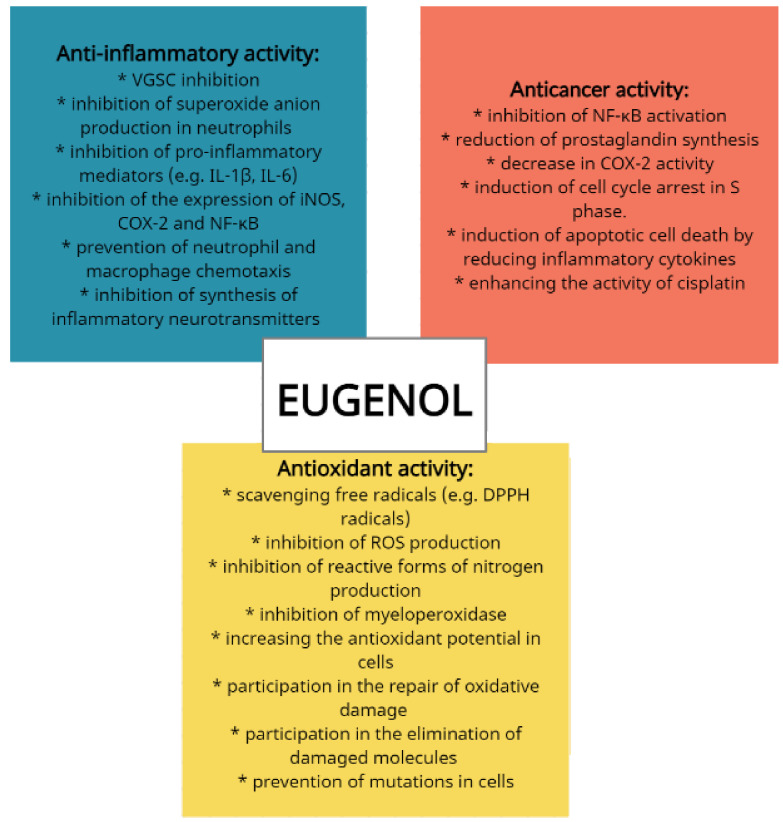
The anticancer, anti-inflammatory, and antioxidant mechanism of action of eugenol (own elaboration based on [1,3,4,7,14,15,16,28,29,30,31,32]).

**Table 1 ijms-22-03671-t001:** Occurrence of eugenol and its concentration in particular parts of plants [10].

Plant	Part	Concentration (mg/g)
Clove, Clovetree	Flower Leaf, stem	1809
Clover pepper	Fruit	36
Betel Pepper	Leaf	17.85
Carrot	Seed	7
Tulsi	Leaf	4.2–4.97
Ceylon Cinnamon, Cinnamon	Bark	3.52
Turmeric	Leaf, essential oil	2.1
Bay, Bay Laurel	Leaf	1.34
Chinese Ginger	Rhizome	0.4
Nutmeg	Seed	0.32
Small-Flowered Oregano	Shoot	0.055–0.125

**Table 2 ijms-22-03671-t002:** Effects of eugenol in agriculture on various pathogens [9].

Strain	Application	Activity
*Aspergillus ochraceus*	Grains	Inhibition Ochratoxin A synthesis
*Phlyctena vagabunda*, *Penicillium expansum*, *Botrytis cinerea*, *Monilinia fructigena*,	Apples	Fungicidal
*Sclerotinia sclerotiorum, Rhizopus stolonifer*, *Mucor* spp.	Peaches	Fungicidal
*Saccharomyces bayanus, Rhodotorula bacarum,* *Pichia membranifaciens*	Fruit juice	Fungicidal

## Data Availability

Not applicable.

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
