# Peer review of "Biological Properties and Prospects for the Application of Eugenol—A Review"

_ijms, 2021, doi:10.3390/ijms22073671_

Round 1

Reviewer 1 Report

The manuscript by Ulanowska M and Olas B addresses the role and application of biological properties of eugenol-A. This manuscript is presenting interesting data, however the draft has a number of minor points that need to be addressed:

  1. Generally, the biological properties and application of eugenol-A are well-presented supported by relevant references from existing literature.
  2. The authors should comment the purity of eugenol-A following isolation and its biological availability/stability in a short or long time of storage in solvents.
  3. The authors should provide an illustration describing the important targets in signalling pathways affected by eugenol-A and derivatives.
  4. The authors should comment about current or future clinical validation of eugenol-A and derivatives for treatment of diseases.

Author Response

The manuscript by Ulanowska M and Olas B addresses the role and application of biological properties of eugenol-A. This manuscript is presenting interesting data, however the draft has a number of minor points that need to be addressed:

  1. Generally, the biological properties and application of eugenol-A are well-presented supported by relevant references from existing literature.

Response: The biological properties of eugenol are described in different review paper. However, they published in 2010, 2016 and 2018.

  1. The authors should comment the purity of eugenol-A following isolation and its biological availability/stability in a short or long time of storage in solvents.

Response: A section on bioavailability and encapsulation of eugenol has been added.

  1. The authors should provide an illustration describing the important targets in signalling pathways affected by eugenol-A and derivatives.

Response: Antibacterial and antifungal activity of eugenol (own elaboration based on [2,10,20]) has described on Fig. 2. The anticancer, anti-inflammatory and antioxidant mechanism of action of eugenol are shown in the newly added Figure 5.

  1. The authors should comment about current or future clinical validation of eugenol-A and derivatives for treatment of diseases.

Response: We have described this information in Conclusion.

Reviewer 2 Report

This article is a review on the medicinal and agricultural uses of eugenol. Overall, I found the article very repetitive and in need of better organization. I also think it would about the biosynthesis of eugenol.

line 27 "A iso eugenol derivatives" This should be "Isoeugenol derivatives"

line 34 I believe that eugenol is commonly regarded as a phenylpropanoid as it proposed synthesis pathway is from phenylalanine.

line 55 "is not an ideal compound" Is not an ideal compound for what? The rest of the article is about all the medicinal uses for it. So, is it not an ideal compound for medicinal use?

line 62 "mouth," should be "mouth."

lines 60-69 This paragraph seems unnecessary. The sentence in lines 56-59 is sufficient.

The antioxidant activity section needs to be reorganized. Currently, the authors talk about antioxidant properties, then oxidant properties, then antioxidant properties again.

The same should be done with the sections on biofilms. These section should be together.

Lines 185-193 Please write this as sentences or refer to your figure.

Author Response

This article is a review on the medicinal and agricultural uses of eugenol. Overall, I found the article very repetitive and in need of better organization. I also think it would about the biosynthesis of eugenol.

line 27 "A iso eugenol derivatives" This should be "Isoeugenol derivatives"

Response: It has been corrected.

line 34 I believe that eugenol is commonly regarded as a phenylpropanoid as it proposed synthesis pathway is from phenylalanine.

Response: It has been corrected.

line 55 "is not an ideal compound" Is not an ideal compound for what? The rest of the article is about all the medicinal uses for it. So, is it not an ideal compound for medicinal use?

Response:  It has been corrected.

line 62 "mouth," should be "mouth."

Response: The paragraph with that word has been deleted.

lines 60-69 This paragraph seems unnecessary. The sentence in lines 56-59 is sufficient.

Response: This paragraph has been deleted.

The antioxidant activity section needs to be reorganized. Currently, the authors talk about antioxidant properties, then oxidant properties, then antioxidant properties again.

Response: This section has been reorganized, the paragraph on oxidation has been moved to the end.

The same should be done with the sections on biofilms. These section should be together.

Response: This section has been reorganized.

Lines 185-193 Please write this as sentences or refer to your figure.

Response: This part has been written as a sentence.

Round 2

Reviewer 2 Report

The are several very long sentences that make this article difficult to read. For example,

" Eugenol also has a synergistic effect with various antibiotics for example vancomycin, peni-362 cillin and erythromycin - it may increase their effect and reduce their minimum inhibitory 363 concentration (MIC), which may have a significant impact on reducing the antibiotic re-364 sistance of pathogens."

As the sentence structure sometimes makes it difficult to understand. For example,  "Moreover, eugenol significantly affected the bacterial biofilm, because it caused inhibition 187 of biofilm formation, reducing the viability of cells that are part of the biofilm and dis-  188 persing cells within the biofilm matrix, inactivation of biofilm bacterial cells and inhibition 189 of biofilm-associated gene expression (for example pgaA gene). "

Line 34 This sentence has not been corrected. It now reads "Eugenol, i.e. C10H12O2; phenylpropanoid (Figure. 1) is a phenolic aromatic compound belonging to the terpenes." This indicates it is a phenylpropanoid and a terpene. Please read some articles about the biosynthesis of eugenol.

Author Response

The are several very long sentences that make this article difficult to read. For example,

" Eugenol also has a synergistic effect with various antibiotics, for example vancomycin, peni-362 cillin and erythromycin - it may increase their effect and reduce their minimum inhibitory 363 concentration (MIC), which may have a significant impact on reducing the antibiotic re-364 sistance of pathogens."

Response: It has been corrected.

As the sentence structure sometimes makes it difficult to understand. For example,  "Moreover, eugenol significantly affected the bacterial biofilm, because it caused inhibition 187 of biofilm formation, reducing the viability of cells that are part of the biofilm and dis-  188 persing cells within the biofilm matrix, inactivation of biofilm bacterial cells and inhibition 189 of biofilm-associated gene expression (for example pgaA gene). "

Response: It has been corrected.

Line 34 This sentence has not been corrected. It now reads "Eugenol, i.e. C10H12O2; phenylpropanoid (Figure. 1) is a phenolic aromatic compound belonging to the terpenes." This indicates it is a phenylpropanoid and a terpene. Please read some articles about the biosynthesis of eugenol.

Response: It has been corrected.

Round 3

Reviewer 2 Report

 Recommended edit lines 42-49

Eugenol can also be produced synthetically by allylation of guaiacol with allyl  chloride, and biotechnologically by the biotransformation of a wide range of microorganisms, such as Corynebacterium spp., Streptomyces spp. and Escherichia coli [2,7-9]. After extraction, eugenol appears as a clear to pale yellow liquid with an oily consistency and a spicy aroma. It is sparingly soluble in water and well soluble in organic solvents. Orally administered, eugenol is rapidly absorbed and metabolized in the liver. As an essential oil, it is rapidly absorbed through the stomach and skin, among other organs. 

Lines 187-190 belong with the paragraph on biofilm starting line 164

Line 240 rewrite "cyclooxygenase  II  enzymes. In  addition,"

Author Response

Recommended edit lines 42-49

Eugenol can also be produced synthetically by allylation of guaiacol with allyl  chloride, and biotechnologically by the biotransformation of a wide range of microorganisms, such as Corynebacterium spp., Streptomyces spp. and Escherichia coli [2,7-9]. After extraction, eugenol appears as a clear to pale yellow liquid with an oily consistency and a spicy aroma. It is sparingly soluble in water and well soluble in organic solvents. Orally administered, eugenol is rapidly absorbed and metabolized in the liver. As an essential oil, it is rapidly absorbed through the stomach and skin, among other organs. 

Response: It has been corrected.

Lines 187-190 belong with the paragraph on biofilm starting line 164

Response: It has been corrected.

Line 240 rewrite "cyclooxygenase  II  enzymes. In  addition,"

Response: It has been corrected.